# Direct and Indirect Contributions of Three Aspects of Morphological Knowledge to Second Language Reading Comprehension

**Junko Yamashita [1],* and Kunihiro Kusanagi [2]**

1   Graduate School of Humanities, Nagoya University, Nagoya 464-8601, Japan
2   Faculty of Regional Development, Prefectural University of Hiroshima, Hiroshima 734-8558, Japan; kusanagi.kuni@gmail.com
*   Correspondence: yamashita@nagoya-u.jp

**Abstract:** Growing attention has been devoted to the contribution of morphological knowledge to reading comprehension. Because of the complex nature of morphological knowledge, more fine-grained approaches are sought on this topic by exploring multiple aspects of morphological knowledge and multiple pathways through which each aspect contributes to reading comprehension. This study measured three aspects of affix knowledge (form, meaning, and use) and vocabulary breadth and examined how each aspect contributes to EFL (English as a foreign language) reading comprehension by modeling direct and indirect effects with vocabulary as a mediator. The participants were 211 Japanese university students. All variables were measured using standardized tests. Direct effects of meaning, use, and vocabulary and indirect effects of meaning and use via vocabulary were significant. However, form displayed no significant effect. The lack of significant effects for form may be due to the design of this study, which did not include word reading (a variable that may mediate form's effect). In sum, although the form aspect did not show any effect, semantic and syntactic aspects demonstrated direct and indirect contributions. Overall, this study endorsed the criticality of a more fine-grained approach, shedding light on what and how morphological knowledge supports L2 reading comprehension.

**Keywords:** L2 reading comprehension; morphological knowledge; vocabulary; multiple aspects; direct path; indirect path; path analysis





## 1. Introduction

Reading comprehension is a complex cognitive skill supported by multiple components [1]. Morphological knowledge, the focus of this study, has been recognized as one critical component [2,3], especially for more advanced-level readers who read academic texts involving increasing levels of morphologically complex vocabulary [4]. Second language (L2) reading research has witnessed a rapid increase in investigation into morphological knowledge in the componential approach to reading. Illustrative of this trend is the change in the number of samples in meta-analyses. According to their inclusion criteria, Jeon and Yamashita [5] included only six samples found in research covering a span of about 32 years (January 1979 to May 2011), while the updated version [1] using largely identical criteria added eight new samples found in the following six years (June 2011 to July 2017) (in total, 14 samples). This change indicates a 133% increase. Although the number of studies was still smaller than other well-researched components such as vocabulary and first language (L1) reading, morphological knowledge documented the most prominent rate of increase among the 11 components examined in [1].

Due to the intricate nature of morphological knowledge, various terminologies have been used to refer to knowledge and skills readers acquire in relation to morphology, such as morphological awareness, morphological knowledge, morphological processing,

morphological skills, and morphological analysis. Distinctions among terms are "quite murky" ([6], p. 266). For this paper, we use "morphological knowledge" as a general term (see 1 and 4). This study focuses on the knowledge of derivational morphology (affixes) in English, and the method section elaborates on how we operationalize this type of knowledge.

In L1 readers, morphological knowledge gradually develops through elementary school (e.g., [7–11]) and secondary school up to university [12,13]. L2 studies, although smaller in number, have also documented the development of morphological knowledge longitudinally [14,15] or cross-sectionally [16] from secondary- to tertiary-level learners. Several meta-analyses have supported the effectiveness of morphological instruction in enhancing morphological knowledge both in L1 and L2, but it remains inconclusive as to whether morphological instruction improves reading comprehension [17–19].

### 1.1. How Morphological Knowledge Supports Reading Comprehension

Morphology interacts with other linguistic systems because morphemes embody phonological/orthographic forms, semantic meanings, and syntactic functions. Therefore, the effects of morphological knowledge on reading comprehension can easily become spurious because of the overlap of variance with other components. Researchers often include various covariates such as phonological awareness, vocabulary, and word reading in their analyses to partial out the effects of related variables and identify the unique contribution of morphological knowledge to reading skills. A large body of research has supported unique contributions of morphological knowledge to reading comprehension from children to adults in both L1 and L2 readers (e.g., L1: [7–10,20–26] and L2: [27–31]). However, some studies did not find unique effects in the presence of other variables (L1 and L2: [32–34]), while others found different results across different reader groups (L1: [35] and L2: [36]) or different types of morphological knowledge (L2: [37,38]). Overall, however, the field has agreed on the contribution of morphological knowledge to reading comprehension. Meta-analyses on the relationship between morphological knowledge and reading comprehension are informative as a summary of past L2 studies. Jeon and Yamashita [1] reported a large effect size ($r = 0.635$), which was larger than those of other lower-level components examined in their study (decoding [0.586], phonological awareness [0.611], and orthographic knowledge [0.590]). Another meta-analysis with L2 children reported a strong effect size ($r = 0.52$) as well [39].

Based on the consensus on the importance of morphological knowledge, more recent studies have been expanding the scope of the investigation. The dimensionality of morphological knowledge and the multiple ways it may contribute to reading comprehension have been particular subjects of attention [40]. Theoretical advancement is also evident, as shown by the Morphological Pathways Framework [41]. It is built upon the Reading Systems Framework [42] and expands it to specify the roles of morphology in text comprehension. Before explaining this model further, it is worth noting that this is a model for L1 children. Therefore, morphological awareness is defined as "the ability to reflect on and manipulate morphemes in *spoken* language" [41] (p. 12, emphasis added). Although it is not uncommon for morphological awareness to be seen as a property of oral language, some L1 reading researchers do not wish to restrict the concept to spoken language [3]. This broader view, accommodating both oral and written language, is important for L2 research because L2 learners often learn spoken and written forms of L2 simultaneously, and morphological knowledge is typically measured with written tests. We, therefore, do not differentiate language mode (oral vs. written) in our conceptualization of morphological knowledge.

The Morphological Pathways Framework [41] postulates multiple pathways (one direct and two indirect) connecting morphology to reading comprehension. The direct path depicts the direct effect of morphological awareness on text comprehension, representing the unique contribution of morphological awareness found in previous studies. The two indirect pathways are (1) via lexicon (vocabulary) through morphological analysis (analyzing word meanings or lexical inferencing using morphemes) and (2) via word

reading through morphological decoding (reading a word by utilizing morphological decomposition). The idea of these direct and indirect pathways is not unique to this model (e.g., [3]); however, the Morphological Pathways Framework provided "the first testable model" [40] (p. 2) to explore the role of morphology in literacy acquisition.

The literature has increasingly examined direct and indirect pathways from morphological knowledge to reading comprehension, but the results are mixed. In L1 research, [43] studied English-speaking children in grade 3. The direct path and indirect path via word reading were significant, but the indirect path via vocabulary was insignificant. Many L2 studies have found the direct path to be significant as well [15,43–50], but [51] and [52] did not. Furthermore, many found a significant indirect path via vocabulary [15,44,46–49,51,52], but [45,50] did not. Ahang and Koda [52] tested indirect paths via lexical inferencing as well as via vocabulary and found both were significant. Results regarding the indirect path via word reading are even more mixed; some studies found it significant [49,50], but others did not [45,48,49,51]. Goodwin et al. [45] tested three mediators (listening comprehension, vocabulary, and word reading) and found that only listening comprehension served as a mediator. Zhang [15] and Qio et al. [50] tested indirect contributions with vocabulary and word reading as mediators in sequence. In Zhang [15]'s longitudinal study, morphological knowledge at Time 1 affected reading comprehension at Time 2 in the following sequence: morphological knowledge [Time 1] → word reading [Time 1] → vocabulary [Time 1] → reading comprehension [Time 2]. On the other hand, the sequence Qiao et al. [50] found was as follows: morphological knowledge → vocabulary → word reading → reading comprehension. The contrast in the sequential mediation of vocabulary and word reading in these studies is interesting, as both examined L1 Chinese children at around the same grade levels (grade 3 vs. grades 3 and 4). However, these studies differ in many aspects, including the research design, measures, and sociocultural/educational contexts.

*1.2. Multiple Aspects of Morphological Knowledge*

Like any component of reading, morphological knowledge is multifaceted. A considerable variety exists in measures of morphological knowledge, potentially tapping into different aspects of morphological knowledge. Measures differ in many respects, including types of morphemes (inflection/derivation/compound), test tasks (production/recognition, spoken/written, and with/without context), and test item types (real words/pseudowords). Regarding the test item type, researchers are aware that real word items are more likely to tap into vocabulary knowledge [3,35] and that pseudoword items are more likely to encourage test-takers to focus on morphemes [8]. However, real word items are also popular in tests of morphological knowledge (e.g., [9,12]), and some researchers recommend the use of both real words and pseudowords in the measures depending on how the construct of morphological knowledge is perceived or the age range of participants [35,53].

Although the various measures reflect the complex nature of morphological knowledge, researchers in the field have initiated efforts to understand the multiple dimensions of morphological knowledge at more abstract levels. Empirical investigations into the dimensionality have emerged in L1 research. A typical approach is to measure the construct with various tests and apply factor analysis or related statistical methods to extract underlying factors (cf. Yopp [54], a seminal work for phonemic awareness). James et al. [35] gave six morphological tests to three groups of English-speaking children (ages 6–8, 9–11, and 12–13). Principal component analyses yielded a single factor in all groups, which suggests uni-dimensionality. However, many studies have identified multiple factors. For example, Tighe and Schatschneider [55] administered seven morphological tests and two vocabulary tests to English-speaking adults in adult basic education and examined dimensionality with confirmatory factor analysis. When the nine tests were analyzed together, three factors emerged: real word morphology (tests using real words were loaded), pseudoword morphology (tests using pseudowords were loaded), and vocabulary. Despite a high correlation between real word morphology and pseudoword morphology (0.94), the three-factor model fitted the data better than a two-factor model consisting of morphology and vocabulary fac-

tors. Tighe and Schatschneider [55] argued that adults may use morphological knowledge differently when they read real words and pseudowords. The vocabulary factor correlated more highly with the real word morphology (0.68) than with pseudoword morphology (0.50), which would be reasonable and resonate with some researchers' views [3,35].

Goodwin and colleagues [5,56] have been extensively investigating the dimensionality and support the multidimensionality of morphological knowledge. Goodwin et al. [5] gave seven morphological tasks/tests to English-speaking children in grades 7 and 8. A bifactor model fitted the data well, indicating general morphological knowledge and seven task-specific variances. Goodwin et al. [56] is a large-scale study involving 3214 English-speaking children from grades 5 to 8 (8% were L2 English speakers). They administered 14 morphological tasks/tests and included 10 in their analysis (tasks with weak psychometric properties, such as the ceiling effect, were dropped). The results supported multidimensionality, consisting of four skills factors (morphological awareness, morphological–syntactic knowledge, morphological–semantic knowledge, and morphological–orthographic/phonological knowledge) and 10 task-specific factors. These L1 studies suggest that the investigation into the dimensionality of morphological knowledge has progressed, but we still need further investigations to arrive at a consensus.

L2 researchers are also aware of the multi-faceted nature of morphological knowledge, but many have made theoretical distinctions rather than taking empirical approaches to the dimensions. Nation [57]'s framework is influential in conceptualizing different aspects of word knowledge in L2 research. He proposes three dimensions at the most general level (form, meaning, and use). Applying this framework to derivational morphology, Sasao and Webb [58] created the Word Part Levels Test (WPLT), "a comprehensive measure of affix knowledge" (p. 14), to provide diagnostic information on learners' strengths and weaknesses of affix knowledge. In this test, form means the orthographic form (spelling) of morphemes, as the WPLT is a written test; meaning refers to the semantic meaning; and use indicates the syntactic function or parts of speech. It is noteworthy that these three aspects coincide with the dimensions identified by Goodwin et al. [56]'s modeling study cited above (i.e., syntactic, semantic, and orthographic/phonological (i.e., form) knowledge). The distinction among these three aspects of knowledge was succeeded by the Computer Adaptive Testing version of the WPLT (CAT-WPLT) by Mizumoto et al. [59]. Yet another theoretical distinction was adopted by Alshehri and Zhang [27]: knowledge (accuracy) and efficiency (speed).

To summarize, the multi-faceted nature of morphological knowledge has been recognized in both L1 and L2 research. Although no consensus has yet arisen, researchers are utilizing conceptual and empirical approaches to understand the dimensionality of morphological knowledge, aligning with their own research purposes.

*1.3. Contributions of Different Aspects of Morphological Knowledge to Reading Comprehension*

By combining insights from the multifaceted view of morphological knowledge and the multiple ways it may contribute to reading, we can ask a more fine-grained question: which aspects of morphological knowledge contribute to reading comprehension in what ways? Researchers have elaborated on these insights both individually and together. As reviewed above, Goodwin et al. [5] found a general factor and seven task-specific factors of morphological knowledge. They also tested the direct contributions of these eight factors to reading comprehension. The general factor made the most extensive contributions, and the morphological meaning factor (self-assessed knowledge of the meaning of the base word and its derived word) made a more minor but significant contribution. On the other hand, the morphological spelling factor (spelling a dictated derived word) made a negative contribution; Goodwin et al. [5] speculated that too much attention to spelling might hinder comprehension. Levesque et al. [43] tested direct and indirect contributions of morphological awareness to reading comprehension by postulating four mediators (morphological decoding, morphological analysis, vocabulary, and word reading) with L1 English children in grade 3. Three paths from morphological awareness to reading

comprehension were significant: the direct path and two indirect paths (via morphological analysis and via sequential mediation of morphological decoding and word reading (i.e., morphological awareness → morphological decoding → word reading → reading comprehension)). Unexpectedly, morphological awareness contributed to vocabulary, but the path from vocabulary to reading comprehension was not significant beyond mediators included in the analysis (despite a significant correlation between vocabulary and reading comprehension); in other words, vocabulary was not a mediator. The authors argued that the mediating role of vocabulary may appear in older children and called for further studies. Levesque et al. [60] is a longitudinal study that followed L1 English children from grade 3 to grade 4. They measured morphological awareness and morphological analysis (inferring the meaning of unfamiliar derived words). Morphological analysis predicted gains in reading comprehension, and morphological awareness predicted gains in morphological analysis. In other words, morphological awareness contributed indirectly to reading comprehension via morphological analysis.

Only a few studies have documented the relationship between different aspects of morphological knowledge and reading comprehension in L2 learners (but none combined multiple aspects with multiple pathways). Zhang and Koda [37] found that knowledge of derivation and compounding, but not that of inflection, contributed to reading comprehension in L1 Chinese/L2 English children. Alshehri and Zhang [27] adopted the distinction between the knowledge and processing of derivational morphemes with L1 Arabic/L2 English university students. Both aspects contributed to reading comprehension, but knowledge showed stronger effects than processing.

### 1.4. The Present Study

The present study examined the contributions of morphological knowledge to L2 reading comprehension. More specifically, we focused on derivational morphemes (or affixes) in English as a foreign language. In line with the recent expansions of the research scope of this topic, as reviewed above, we elaborated on the multiple aspects and multiple contribution pathways of affix knowledge. Like Sasao and Webb [58] and Mizumoto et al. [59], we adopted the theoretical distinction of three aspects of knowledge (form, meaning, and use) because this conceptualization is founded on the well-accepted comprehensive framework of vocabulary knowledge in L2 research [57]. Also note that, as mentioned above, a large-scale L1 study supports the distinction of these aspects [56]. As for the multiple pathways, we tested the direct and indirect contributions of each aspect of affix knowledge to reading comprehension. The mediator in the indirect path was vocabulary breadth. While most of the research on morphological knowledge and reading comprehension has been conducted with children both in the L1 and L2 fields, this study added data from tertiary-level students (L1 Japanese university students in an EFL context). The study did not investigate cross-linguistic influence (e.g., [30,38,61]). However, by involving Japanese as the only L1, we controlled for potential contamination in the results that may arise from different L1s. The following research questions guided this study:

1. Do form, meaning, and use of morphological knowledge and vocabulary make direct contributions to L2 reading comprehension?
2. Do form, meaning, and use of morphological knowledge make indirect contributions to L2 reading comprehension via vocabulary?

## 2. Method

### 2.1. Participants

L1 Japanese university students at a university in Japan participated in this study ($N$ = 211, 141 males and 70 females). English is not a societal language in Japan, and students learn it as a foreign language in school. Before entering university, the participants had learned English for at least six years during their secondary school education. The participants had diverse fields of study including economics, literature, law, engineering, science, agriculture, medicine, and informatics. Their estimated overall mean TOEIC

(Test of English for International Communication) score (see below) was 665.75 (m), with means of 349.65 (*SD* = 58.01) and 316.16 (*SD* = 56.08) in the listening and reading sections respectively. These scores suggest average English proficiency at the B1 level according to the Common European Framework of Reference for Languages (CEFR) (ranging from upper A2 to C1) [62].

*2.2. Measures*

2.2.1. Affix Knowledge

To measure the three aspects of affix knowledge (form, meaning, and use), an online test, the Computer Adaptive Testing version of the Word Parts Levels Test (CAT-WPLT) [59], was used. It is available for free on the Internet and can be accessed by individuals. The original paper-based WPLT [58] was developed based on a large sample of participants (*N* = 1348) from over 100 countries (including L1 and L2 English speakers) with high internal consistencies (Cronbach's alpha) in all three sections (0.91, 0.94, and 0.93). Sasao and Webb [58] argue that because of the involvement of a wide range of L1s in the sample, the effects of cognate and loan words of any specific L1 are likely to be negligible. Mizumoto et al. [59] applied the IRT (item response theory) procedure to create the CAT version and demonstrated the equivalent level of measurement precision as the original WPLT using data from 760 Japanese university students. Because items in the computer adaptive tests are individually tailored, tests can be completed with fewer items (thus in shorter times). In the CAT-WPLT, the number of items is 20, 15, and 10 in the form, meaning, and use sections, respectively.

The test has three sections (form, meaning, and use); all take a four-option multiple-choice format. The form section tests the recognition of morphemes (test-takers choose word parts with semantic or syntactic meaning). Distractors are non-morphemic letter strings in English words (e.g., 1. -ing, 2. -nge, 3. -eld, 4. -kle). The meaning section tests semantic knowledge of affixes. An affix is presented with two real word examples (e.g., -ed [walked; played]) to help test-takers understand the tested meaning of affixes. The answer options are listed below the target affix (1. past, 2. not, 3. many, and 4. person). The actual word examples are vital to disambiguate the target meaning in morphemes with multiple senses (e.g., re-, dis-). The use section tests knowledge of parts of speech of suffixes: noun, verb, adjective, and adverb. Since this section necessitates metalinguistic knowledge (the meaning of these grammatical classes), the section first lists two examples for each (e.g., noun: house [My house is old.]; water [They drink water.]) to remind students of the syntactic function of each part of speech. The test format is identical to the meaning section (e.g., -ency [tendency; dependency]), but answer options are always four parts of speech (1. noun, 2. verb, 3. adjective, and 4. adverb).

After completing the test, the system automatically evaluates the test-taker's responses. It then provides diagnostic feedback on the test-taker's level of proficiency (beginner, intermediate, or advanced) in each of the three sections. Test-takers can access the test items via a link on the web page to study affixes based on their individual needs [59]. Although ability scores are not typically displayed to test-takers, the authors made a special arrangement with the test developer and obtained scores from the system's storage.

2.2.2. Reading Comprehension

The reading comprehension sub-section of the VELC (Visualizing English Language Competency) test was used. The VELC test is an English proficiency test for university students in Japan developed by the VELC Research Group with initial data from over 5000 Japanese university students [63]. It is administered on an institutional basis and not available to individuals. It consists of listening and reading sections and estimates the TOEIC score as a guideline for general English proficiency (the current participants' English proficiency reported in the participant section was evaluated in this way) [64]. Since its launch in 2012, many universities have used this test in their curriculums, and the test quality has been constantly monitored and publicized by the research group (e.g., [63–66]).

It shows high test quality: external validity (the high correlation [0.825] with TOEIC score [63]), reliability (over 0.86), and person separation at different levels of university [66]. The reading comprehension sub-section's reliability is also adequate: 0.78 [63] and 0.78 [65]. The test has paper-based and computer-based versions; this study used the computer (online) version.

The test utilizes a multiple-choice gap-filling format. Test-takers choose an answer out of four options that best fills the gap in a text. The source texts are relatively short, ranging from one sentence to one paragraph. There is only one gap in each text, and there are 20 items. Thus, although each source text may be relatively short, test-takers read 20 texts altogether. Even if the text is only a sentence, it has at least two clauses, and the test requires test-takers to understand the semantic content of each clause and choose the option that logically connects them.

### 2.2.3. Vocabulary Breadth

Vocabulary breadth was also measured with the computer version of the VELC test using its vocabulary sub-section in the reading section. Testing experts reported high reliability for this section as well: 0.75 [63] and 0.81 [65]. It is a form recognition test. The item is given with two Japanese words/synonyms, and test-takers choose the English equivalent out of four options (e.g., society, experience, notice, or language). There are 20 items.

The system automatically scores responses to test items. The results are electronically returned to both the institution and the test-taker. While the institution can access the full results, each test-taker can only see their own result. This result includes the test-takers' scores, feedback statements on their strengths and weaknesses in each subsection, and personalized study advice.

### 2.3. Procedure

The tests were administered as part of coursework in a participants' EFL class. Regardless of their achievement levels, they received a 10% bonus in their course evaluation after taking the tests and answering a questionnaire about each test. The educational benefit of the tests is for the students to obtain diagnostic statements on their English ability. In addition, the CAT-WPLT provides an affix list, and the VELC test offers study advice (cf. the Material section in this paper). The first author explained both the research and educational purposes of the tests, and the participants understood and signed the consent form before taking the tests.

The participants first took the CAT-WPLT individually in their free time. All students completed the test within a week. The test has no time limit but is estimated to take about 10 min to complete [59]. Motivated to receive an accurate diagnosis, the students refrained from using external resources such as a dictionary or online translator and completed the test without any issues. This was confirmed by the questionnaire they filled in after the test. After a few weeks, the VELC test was administered in a group of about 20 to 40 students in the university's computer lab with the attendance of the first author or a research assistant. Each student was seated at a desktop computer and took the test by themselves, following the instructions on the computer. Although this study used scores from only the vocabulary and reading comprehension sub-sections, the students took the whole test (70 min) as it is the requirement of the test administration and necessary to receive diagnostic feedback.

### 3. Results

Table 1 summarizes descriptive statistics of the five variables. The VELC test returns standardized scores with a score of 500 as the mean based on the past data of Japanese students. The mean scores of 650.53 and 605.86 in the reading comprehension and vocabulary sub-sections show that the means of the current participants were higher than the average of the large sample of Japanese students in the database. The CAT-WPLT returns theta scores expressed in logits computed by the IRT procedure. A higher score means

a higher ability level. As information to compute reliability coefficients is not available from either test in the context of this study, sample-specific reliability coefficients were not computed. However, based on the previous large-scale studies (with Japanese university students) reviewed above, we assume sufficient reliability of these tests for the current Japanese participants.

**Table 1.** Descriptive statistics of morphological knowledge, vocabulary, and reading comprehension.

| | Minimum | Maximum | *M* | *SD* | Skewness | Kurtosis |
|---|---|---|---|---|---|---|
| Reading Comprehension | 430.00 | 814.00 | 650.53 | 89.93 | 0.31 | −0.66 |
| Vocabulary | 417.00 | 707.00 | 605.86 | 70.13 | −0.17 | −0.66 |
| Form | −2.51 | 1.61 | 0.01 | 0.59 | −0.19 | 2.26 |
| Meaning | −2.30 | 1.15 | −0.25 | 0.53 | 0.01 | 0.87 |
| Use | −2.35 | 1.28 | 0.17 | 0.70 | −0.97 | 1.19 |

Correlation coefficients are shown in Table 2. All correlations were significant, but they were in the range of low to moderate, which eliminated concerns about multicollinearity. The three aspects of morphological knowledge correlated moderately with each other. Reading comprehension correlated with all four predictors to modest degrees, and vocabulary correlated with all three aspects of morphological knowledge from low to modest degrees. Among the three aspects of morphological knowledge, form correlated with reading comprehension and vocabulary most weakly.

**Table 2.** Correlations among all variables.

| | 1. Reading Comprehension | 2. Vocabulary | 3. Form | 4. Meaning | 5. Use |
|---|---|---|---|---|---|
| 1 | 1.000 | 0.259 ** | 0.338 ** | 0.356 ** | 0.422 ** |
| 2 | | 1.000 | 0.161 * | 0.221 ** | 0.213 ** |
| 3 | | | 1.000 | 0.367 ** | 0.424 ** |
| 4 | | | | 1.000 | 0.347 ** |
| 5 | | | | | 1.000 |

*Note*: * $p < 0.05$ and ** $p < 0.01$.

A series of path analyses was conducted to answer the research questions. Aligning with theoretical expectations, the model initially assumed direct contributions of all predictors and indirect contributions (via vocabulary) of each morphemic predictor (form, meaning, and use) to reading comprehension. Our modeling procedure resulted in the removal of the direct and indirect contributions of form because the regression coefficients did not reach statistical significance and the variable's inclusion severely decreased the goodness of fit. After removing the path, the final model showed a good fit to the data ($X^2(2) = 4.142$, $p = 0.126$, CFI = 0.986, TLI = 0.930, RMSEA = 0.071 [0.071–0.170], SRMR = 0.029), explaining 25% of the variance in the reading comprehension score (Figure 1). Details of the results are summarized in Table 3. Three direct effects on reading comprehension were significant: vocabulary (β = 0.144), meaning (0.214), and use (0.317). Meaning and use also had significant direct effects on vocabulary (meaning: 0.167; use: 0.155), which led to significant indirect effects (via vocabulary) on reading comprehension (meaning: 0.024; use: 0.022). Altogether, the total effects of meaning and use on reading comprehension were 0.250 and 0.366, respectively. In contrast, there was no significant effect of form either on vocabulary or reading comprehension. Statistical analyses were conducted with R version 4.0.5 [67] using the maximum likelihood estimation of Wishart parameters.

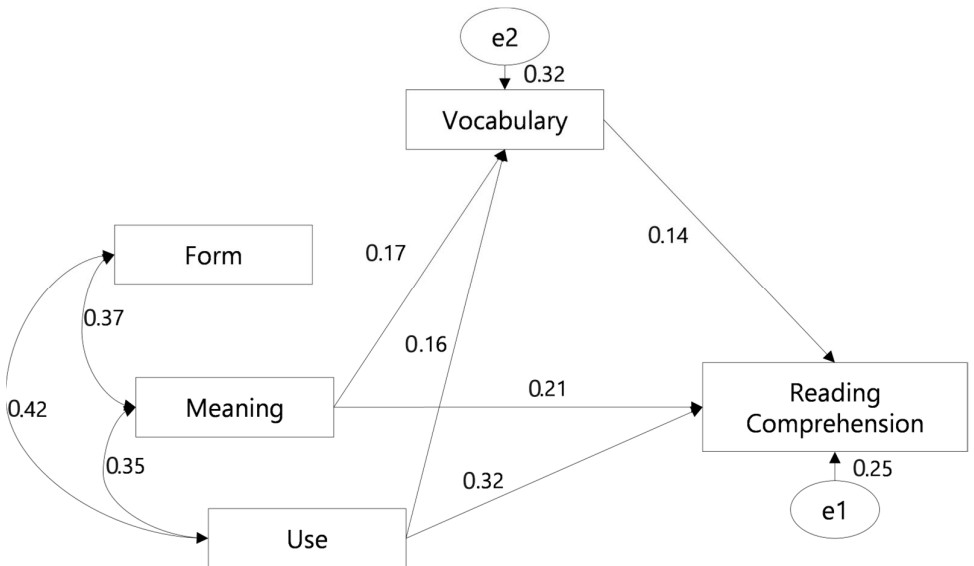

**Figure 1.** Path diagram of the final model.

**Table 3.** Summary of path coefficients in the final model.

| | Estimate | *SE* | *p* | Standardized |
|---|---|---|---|---|
| Regression | | | | |
| Reading from Vocabulary | 0.185 | 0.080 | 0.020 | 0.144 |
| Reading from Meaning | 36.368 | 10.962 | 0.001 | 0.214 |
| Reading from Use | 40.810 | 8.318 | <0.001 | 0.317 |
| Vocabulary from Meaning | 22.028 | 9.381 | 0.019 | 0.167 |
| Vocabulary from Use | 15.598 | 7.131 | 0.029 | 0.155 |
| Covariances | | | | |
| Meaning and Form | 0.114 | 0.023 | <0.001 | 0.367 |
| Meaning and Use | 0.129 | 0.027 | <0.001 | 0.347 |
| Form and Use | 0.174 | 0.031 | <0.001 | 0.424 |
| Total Effects | | | | |
| Meaning | 837.500 | 424.409 | 0.048 | 0.250 |
| Use | 677.363 | 322.105 | 0.035 | 0.366 |

## 4. Discussion

This study investigated how three aspects of affix knowledge and vocabulary breadth contribute to the L2 reading comprehension of tertiary-level EFL learners, modeling the direct effects of all predictor variables and the indirect effects of the three aspects of affix knowledge via vocabulary. The three aspects of affix knowledge were form (ability to recognize the orthographic form of morphemes), meaning (ability to recognize the semantic meaning of morphemes), and use (ability to recognize the parts of speech of morphemes).

Correlations among the three aspects of affix knowledge were significant and moderate (0.347 to 0.424), suggesting that, although some common knowledge/ability underlies these aspects, their constructs are not identical. Mizumoto et al. [59] also found moderate correlations among the three aspects among Japanese students (from 0.337 to 0.527) and supported the value of measuring the three aspects of affix knowledge and reporting separate scores rather than a total single score, especially for diagnostic purposes (separate scores more specifically show weaknesses of individual students). In their results, the correlation between use and form was the lowest; Mizumoto et al. [59] argued that this may be because their participants lacked metalinguistic knowledge of the parts of speech. This study, however, did not replicate this pattern of correlations. Mizumoto et al. [59]'s

view on metalinguistic knowledge did not apply to the current participants, perhaps due to the relatively higher overall L2 proficiency of the current participants (A2 to B1 in [59] and A2 to C1 in this study).

Our research questions focused on the direct and indirect contributions of component variables to reading comprehension. We found that meaning, use, and vocabulary showed significant direct effects on reading comprehension. The two morphological variables also contributed to vocabulary, through which they made indirect contributions to reading comprehension. The contribution of affix knowledge to vocabulary breadth endorses the associations between affix knowledge and vocabulary knowledge [14,34,68]. The difference in the magnitude of the effects was not very large: 0.167 (meaning) and 0.155 (use). Thus, both aspects contributed to vocabulary knowledge, which is reasonable because vocabulary knowledge is multifaceted and involves semantic and syntactic aspects [51,57]. The relatively similar degrees of effects on vocabulary made the indirect effects of use and meaning on reading comprehension similar (use = 0.022 and meaning = 0.024). However, use showed a larger total effect on reading comprehension than meaning (0.366 vs. 0.250). Because the total effect is the sum of the direct and indirect effects, the difference was primarily due to the larger direct effect of use (0.344) than meaning (0.210).

The more prominent direct effect of use may seem surprising, given that comprehension is primarily meaning based. Knowledge of the syntactic functions of affixes may help in parsing complex sentences, but comprehension would be difficult if many words are unknown. Rather than contrasting semantic and syntactic knowledge of affixes, a different explanation may be possible for the greater contribution of use. That is, the score in the use section may reflect not only affix knowledge but also higher overall L2 proficiency. As Mizumoto et al. [59] stated, the use section requires metalinguistic knowledge of the parts of speech (grammatical terminologies and their syntactic meanings), which is an extra metalinguistic requirement to answer this section correctly. Students with high metalinguistic knowledge of this type may have higher levels of metalinguistic awareness in other aspects of the language, which is likely to indicate higher levels of L2 proficiency.

Earlier, we reviewed L1 studies that tested the contributions of different aspects of morphological knowledge to reading comprehension (and vocabulary). Here, we focus on findings that are relevant to this study. Studies that explored the meaning aspect of affixes supported the effects on reading comprehension [5,43,60], which is consistent with our study. However, [5] did not find any significant contributions of the syntactic aspect. They used two suffix choice tests, one using real words and the other using pseudowords (students completed a sentence by choosing the answer, which necessitates using grammatical knowledge of suffixes, e.g., "Our teacher taught us how to [jittling, jittles, jittle]", p. 99). Factors that were indicated by these tests did not contribute to reading comprehension or vocabulary. Although both [5] and the current study sought to measure participants' syntactic knowledge of morphemes, considerable differences in test tasks may have resulted in different constructs being measured. Differences in participants' characteristics (e.g., age or language background) may have also contributed to the different results. We certainly need more studies.

Unlike meaning and use, form did not show any significant effects. However, form significantly correlated with reading comprehension. Therefore, the ability to correctly recognize orthographic forms of affixes is associated with reading comprehension. The result only suggests that form is a weaker predictor of reading comprehension than the other two aspects and vocabulary. Although the weaker role of form may sound reasonable, given that it does not include meaning processing (semantic or syntactic meaning) and that the participants had at least six years of English education, the insignificant result may be partially due to the model tested in this study. Based on the findings of previous studies (e.g., [15,44,47–49,51,52]), we set vocabulary as the sole mediator. If we had included word reading, the indirect effect of form may have appeared because correctly identifying morphemes in a word facilitates reading/recognizing morphologically complex words (morphological decoding). The ability to read a complex word fluently allows more

cognitive resources to be directed to higher-level processes necessary for comprehension. The mediating effect of word reading is postulated in the Morphological Pathways Framework [41] and is supported by several previous studies [43,47,50], though the findings are still mixed [48,49,51]. This remains a question to be pursued in future research.

This study is one of the first in L2 research to take a more fine-grained approach to the contribution of morphological knowledge to reading comprehension by testing multiple aspects of L2 affix knowledge and modeling their direct and indirect effects on L2 reading comprehension. Limitations of this study highlight future directions. The first is the non-inclusion of word reading as a mediator. As discussed above, its mediating effect is worth investigating, especially because previous findings are inconclusive. Since an explicit theoretical framework has recently appeared in the field (the Morphological Pathways Framework) [41], including word reading as well as vocabulary as mediators will contribute to testing the theory. In addition, we can also try more complex mediation pathways in sequence by including multiple mediating variables (see [15,50]). Second, we have adopted form, meaning, and use distinctions well-known in L2 vocabulary research to conceptualize affix knowledge. Although it has theoretical support [57] and methodological advantages [58,59], consensus on morphological dimensionality has yet to be reached. Various terminologies used by different researchers for similar concepts may be a problem to be solved in the future (e.g., morphological analysis, morphological–semantic knowledge, and meaning all refer to the meaning aspect of morphemes). Empirical approaches to the dimensionality of L2 morphological knowledge would be an avenue to advance our understanding, as, to our knowledge, this type of research has not been conducted in L2. Thirdly, there is a wide range of tests/tasks of morphological knowledge. Goodwin and colleagues [5,56] found task-specific factors in addition to more general factors. Their result suggests that different tests/tasks tap into separable constructs within morphological knowledge, indicating the importance of how we measure morphological knowledge. The WPLT and CAT-WPLT have the advantage in this respect, as all three sections use an identical test format, which should mitigate the test format effect. However, we should keep in mind that there is no consensus on morphological knowledge tests, and research results may differ if different tests are used. It may be worth noting that the CAT-WPLT is not commonly used in reading research. This study may be the first to use this test in relation to L2 reading comprehension, although similar studies may appear in the future (personal communication with Dr. Morita). Morphology tests commonly used in L1 research may or may not be suitable for L2 learners due to differences in the language background between L1 and L2 readers. For example, because of the lack of basic L2 morphological awareness measures, Jeon [28] modified a popular morphology test in L1 research, Carlisle's derivation test [9]. However, after substantial adjustments to make the test suitable for L2 participants, the adopted test seemed very different from the original. Using different measures may make generalization difficult, but efforts should be continued to explore measures of morphological knowledge and facilitate communication among researchers. Fourth, the test/task effect can also apply to reading and vocabulary measures. We have chosen standardized tests developed for Japanese university students, but the findings should be verified using different tests.

## 5. Conclusions

Among the three aspects of affix knowledge examined in this study, meaning and use directly and indirectly contributed to L2 reading comprehension, but form did not show any significant effects. Different effects of different aspects support Mizumoto et al. [59]'s contention that researchers should measure and report scores for different aspects of knowledge rather than reporting only a single total morphological knowledge score. This study showed which aspects are more likely to contribute to vocabulary and reading comprehension. Therefore, the result of this study suggests emphasizing meaning and use to enhance L2 learners' vocabulary and reading comprehension by teaching affix knowledge. This more fine-grained approach to the contribution of morphological

knowledge to reading comprehension is still rare in L2 research. We hope this study paves the way for future studies to further explore the dimensionality of L2 morphological knowledge and the multiple pathways of its contribution to L2 literacy acquisition.

**Author Contributions:** J.Y. contributed to the conceptualization, methodology, preliminary analysis, original draft preparation, and funding acquisition; K.K. contributed to the software, main analysis, visualization, and reviewing/editing. All authors have read and agreed to the published version of the manuscript.

**Funding:** This study was funded by the JSPS KAKENHI (Grant No. 18K00737) to J.Y.

**Institutional Review Board Statement:** The study was conducted in accordance with the Declaration of Helsinki and approved by the Ethics Committee of the Graduate School of Humanities, Nagoya University (No. NUHM-21-005; date of approval 6 October 2021).

**Informed Consent Statement:** Informed consent was obtained from all participants involved in the study.

**Data Availability Statement:** Data are available here: https://doi.org/10.18999/2009370.

**Acknowledgments:** The authors are grateful to Atsushi Mizumoto for providing the participants' CAT-WPLT scores from the online database and Tetsuhito Shizuka for responding to our queries about the VELC test.

**Conflicts of Interest:** The authors declare no conflicts of interest.

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
