# Peer review of "Direct and Indirect Contributions of Three Aspects of Morphological Knowledge to Second Language Reading Comprehension"

_education, doi:10.3390/educsci14030270_

Round 1
Reviewer 1 Report
Comments and Suggestions for Authors
The paper explores how different aspects of morphological knowledge (Form, Meaning, and Use) directly and indirectly contribute to reading comprehension in English as a foreign language. The results indicate significant direct effects of Meaning, Use, and vocabulary on reading comprehension. Indirect effects of Meaning and Use via vocabulary were also found to be significant, while Form showed no significant effect.
The study underscores the importance of considering multiple aspects of morphological knowledge and their pathways in understanding L2 reading comprehension, a field which is still relatively understudied. Moreover, it has the merit of focusing on a population (Japanese university students) which typically receives less attention, as the bulk of research on morphological knowledge and reading comprehension has been carried out with children. I would have stressed this point, which, in my view, should also be considered when discussing results (but also when making predictions). I was a bit surprised that there was no discussion of L1 background too. Did the authors have any expectation about this, maybe also related to the different writing system? It could have been worth discussing.
Other points that could be elaborated further are suggested in the file.
Overall, the article merits publication, subject to addressing the aspects highlighted here and in the file.

Author Response
The paper explores how different aspects of morphological knowledge (Form, Meaning, and Use) directly and indirectly contribute to reading comprehension in English as a foreign language. The results indicate significant direct effects of Meaning, Use, and vocabulary on reading comprehension. Indirect effects of Meaning and Use via vocabulary were also found to be significant, while Form showed no significant effect.
The study underscores the importance of considering multiple aspects of morphological knowledge and their pathways in understanding L2 reading comprehension, a field which is still relatively understudied. Moreover, it has the merit of focusing on a population (Japanese university students) which typically receives less attention, as the bulk of research on morphological knowledge and reading comprehension has been carried out with children. I would have stressed this point, which, in my view, should also be considered when discussing results (but also when making predictions). I was a bit surprised that there was no discussion of L1 background too. Did the authors have any expectation about this, maybe also related to the different writing system? It could have been worth discussing.
Other points that could be elaborated further are suggested in the file.
> Thank you for the constructive comment. Our responses are shown by > after each comment.
> We have stressed the unique points of this study in the Present study section. We did not examine L1 influence. To do that, we need a different research design such as comparing derivational morphology and compounding according to L1-L2 differences (some researchers did so with the English-Chinese combination). The contrast in the writing system (opaque vs transparent) does not easily fit the English-Japanese combination. To our knowledge and in this design, we cannot offer specific expectations about L1 effects. However, the merit of this study is that we have only one L1 (Japanese) so the result is not likely to be affected by mixed L1 backgrounds. We included this statement in the Present study section.
Overall, the article merits publication, subject to addressing the aspects highlighted here and in the file.
> Thank you for the comments. Our responses to specific feedback inserted in the PDF file are listed below.
- p.3 Line 145 (real word vs pseudoword factors)
> We agree with the Reviewer. It is [54]’s argument. To make it clear, we added “[54] argued that” to the sentence.
- p. 7 Line 318 (how to calculate theta score)
> We slightly changed a sentence from “(a unit of IRT) to “computed by IRT (Iten Response Theory) procedure”. Researchers typically use a statistical package to obtain logit scores. Mathematical explanation of the IRT procedure is beyond the scope of this paper.
- p.7 Line 331 (Correlations of Form and Meaning are similar)
> Yes, we agree, but it is still a fact that the correlation of Form is the smallest among the three morphological variables.
- p.9 Line 392 (why it is surprising)
> Thank you for this comment. We agree that syntax helps comprehension in the way that the Reviewer mentioned. At the same time, semantics helps readers when they don’t know the syntax. Although we are not evaluating which is more important (syntax or semantics), from our experience, we sometimes see students relying on semantics when they cannot parse sentences (sometimes they use semantics too liberally and result in incorrect comprehension). We are not imposing our view, which is seen in the hedges we used (“may seem surprising”). We think it is fine and would like to wait for the editor’s advice.
- Line 414
> The reviewer suggested that [1]’s task is more likely to find effects of the syntactic aspect (but [1] did not). However, to our knowledge, there is no evidence that suffix choice tests are more likely to show effects than the type of test we used. We cannot make an argument with confidence.
6 Line 424 (age)
Thank you for this insight. Rather than age, we added that the participants learned English for at least six years (this information was added in the Participant section in the revision).
- Line 448 (term problem)
> We are glad to know that the reviewer shares the same kind of concern about a terminological practice in the field. Another reviewer also commented on terminology (though from a different viewpoint). We added a paragraph about terminology in the introduction. Therefore, terminological issues are discussed twice in this paper, as this reviewer suggested (background and discussion sections).
- Line 457 (uncommon test)
> We agree with the reviewer. The WPLT (or CAT-WPLT) is not commonly used in reading research. The last paragraph of this paper discussed limitations of this study. We further elaborated on the measurement issue based on this comment referring more specifically to this and other tests.
Reviewer 2 Report
Comments and Suggestions for Authors
Reviewer’s comments and suggestions:
1. Line 27. "I don’t understand the statement you made. Can you please provide some context
and rephrase it? Specifically, could you explain in what context you intended to say, “This may be because initial research interests were directed at beginning readers, who need to learn to read high-frequency (i.e., not complex) words”?"
2. Line 84 to 115. "I don't understand why the authors kept mentioning the direct and indirect paths to morphological awareness. They didn't explain their point clearly in the paragraphs or at the end of the text. It seemed like an incomplete thought with no clear rationale for being mentioned.”
3. Overall, the paper lacks coherence as the authors present new and unrelated ideas without connecting them to the previous ones.
4. There is no mention of the role of morphological knowledge among second language learners in the literature review, except for an incomplete thought on page 5, lines 214- 221.
5. The literature review does not align with the study's rationale and design.
6. I don't see a need for two separate research questions; they could have been merged and
addressed as direct or indirect paths as one research question.
7. I request that the authors provide demographic information about their participants,
including their language background, as English is not their first language. This would aid in understanding their language learning experiences and profiles.
8. I would like the author/s to add further details on the measures they used to test morphological knowledge, such as the number of items in each category, the scoring and/or coding method used, and the reliability of the measure.
9. Similar comment for the rest of the measures to add further details such as the number of items scoring procedures etc.
The topic of this paper is intriguing and the rationale behind it is strong. However, the writing quality is very poor. In my opinion, a complete rewrite of the literature review and methods section is necessary. I also suggest organizing the results section based on the research question and providing an explanation for including correlational analysis and other analyses but the path analysis. The discussion needs to be rewritten and rearranged accordingly for better clarity and coherence. Any spelling, grammar, and punctuation errors should also be corrected during the process.

N/A
Author Response
> Thank you for the review. Our responses are shown by > after each comment.
Reviewer’s comments and suggestions:
1. Line 27. "I don’t understand the statement you made. Can you please provide some context and rephrase it? Specifically, could you explain in what context you intended to say, “This may be because initial research interests were directed at beginning readers, who need to learn to read high-frequency (i.e., not complex) words”?"
- We rephrased and rewrote this part of the text. The sentence mentioned was deleted in this process.
- Line 84 to 115. "I don't understand why the authors kept mentioning the direct and indirect paths to morphological awareness. They didn't explain their point clearly in the paragraphs or at the end of the text. It seemed like an incomplete thought with no clear rationale for being mentioned.”
> This part continues from the previous paragraph, saying “Based on the consensus on the importance of morphological knowledge, more recent studies have been expanding the scope of the investigation. The dimensionality of morphological knowledge and the multiple ways it may contribute to reading comprehension have been particular subjects of attention [38].”
As we discussed, we are focusing on two research areas: dimensionality and multiple pathways. The part mentioned above relates to the “multiple pathways”. To make the connection clearer, we changed the first sentence of this paragraph from “three pathways” to “multiple pathways”. Three is implied by (one direct and two indirect pathways) that follows “multiple pathways”. In addition, the part mentioned above is in the subsection “1.1. How morphological knowledge supports reading comprehension”. To make the connection clearer, we can change the subtitle to something like “1.1. Multiple pathways (or even Direct and indirect pathways) that morphological knowledge supports reading comprehension.” For this possible change, we would like to ask for editors’ advice.
- Overall, the paper lacks coherence as the authors present new and unrelated ideas without connecting them to the previous ones.
> We did our best to write coherently (and tried so in revision). Other reviewers said “The writing is concise and logical” and “I think the overall writing is good”. We would like to ask for the editors’ feedback.
- There is no mention of the role of morphological knowledge among second language learners in the literature review, except for an incomplete thought on page 5, lines 214- 221.
> The role of morphological knowledge among second language learners is reviewed in the first paragraph in subsection 1.1. At the end of this paragraph, we cited the results of the L2 meta-analysis that reported a large effect size of morphological knowledge on reading comprehension (supporting the role of morphological knowledge among L2 learners). Continuing from this fundamental understanding, the part mentioned above is in the subsection “1.3. Contributions of different aspects of morphological knowledge to reading comprehension”. As the general role of morphological knowledge was discussed in Section 1.1, we focused on different aspects of morphological knowledge in Section 1.3. We would like to consult the editors on this point.
- The literature review does not align with the study's rationale and design.
> We did our best to make the alignment. We would like to consult the editors.
- I don't see a need for two separate research questions; they could have been merged and addressed as direct or indirect paths as one research question.
> We tried to do so but found that the two questions are different not only in direct vs. indirect paths but also in predictors and a mediator. If these are merged, the question will become a long, complex sentence. Two separate questions seem better for the general readership. We would like to ask for the editors’ advice.
- I request that the authors provide demographic information about their participants, including their language background, as English is not their first language. This would aid in understanding their language learning experiences and profiles.
> We added demographic information to the Participants section.
- I would like the author/s to add further details on the measures they used to test morphological knowledge, such as the number of items in each category, the scoring and/or coding method used, and the reliability of the measure.
> We already reported the number of items: “In the CAT-WPLT, the number of items is 20, 15, and 10 in the Form, Meaning, and Use sections, respectively.” The scoring and coding method is not open to the public, but the system automatically scores the test responses. We added this information. Because this is a multiple-choice test, scoring is likely to be dichotomous with the coding of correct or wrong. We added some more information in the Method section. The sample-specific reliability cannot be computed because necessary information (responses to each item) is not provided to the researcher. We explained this in the Result section. Therefore, we reported reliabilities from the test development study (.91, .94, and .93). In the case of standardized tests such as this one, it is not uncommon for researchers to report reliabilities from test manuals. The WPLT does not have a test manual, so we reported reliabilities from the original large-scale study. This is the best we can do, and we believe this is an accepted reporting practice.
- Similar comment for the rest of the measures to add further details such as the number of items scoring procedures etc.
> The number of items for the reading comprehension test and vocabulary test were already reported (20 in both). The scoring and coding method is not open to the public, but the system automatically scores the test responses. We added this information. Because this is a multiple-choice test, scoring is likely dichotomous with the coding of correct or wrong. We reported reliabilities from testing studies because necessary information (responses to each item) to compute sample-specific reliability was not provided to us. As we said above, reporting reliabilities from test manuals (or testing studies) is not uncommon in the case of standardized tests (note that reading comprehension and vocabulary tests are from a standardized test called the VELC test).
The topic of this paper is intriguing and the rationale behind it is strong. However, the writing quality is very poor. In my opinion, a complete rewrite of the literature review and methods section is necessary. I also suggest organizing the results section based on the research question and providing an explanation for including correlational analysis and other analyses but the path analysis. The discussion needs to be rewritten and rearranged accordingly for better clarity and coherence. Any spelling, grammar, and punctuation errors should also be corrected during the process.
> We did our best to write well. As we said above, we received different comments from different reviewers. We would like to wait for the editors’ feedback. The paper was checked by an educated native speaker of English prior to the submission, but we did and will make further efforts to correct the remaining errors.
> Regarding statistical analyses, except path analyses, we reported descriptive statistics and correlations. Path analysis answers research questions, but it is conventional for modeling research to report descriptive statistics and correlations as fundamental statistics. We followed the convention. Both statistics are useful to check fundamental data quality such as distribution and multicollinearity.
Reviewer 3 Report
Comments and Suggestions for Authors
Adopting the Morphological Pathways Framework, this study examined the multi-dimensionality of morphological knowledge, and how these different dimensions contribute to reading comprehension in L2 among a total of 211 Japanese English learners. Using the Word Parts Levels Test, the author found that Meaning and Use contributed to reading comprehension either directly or indirectly via vocabulary knowledge, while there was no relationship between Form and reading comprehension in L2 English. The authors discussed the multi-dimensional nature of morphological knowledge and their relationship with L2 reading comprehension.
Comments:
Overall, the study has a clear problem formulation. It clearly reviews the relevant literature and proposes a problem that needs more research attention. The writing is concise and logical. However, I have a few concerns regarding its definition of key terms and tasks adopted. Please see below for details:
Line 24-29: More citations needed.
Line 64: [3]: I am not sure this is how to do in-text citation in the current citation format. It seems that Author Name [3] more commonly used in other articles published in this journal. The same comment applies to all similar in-text citations in this manuscript.
Line 85-87: Morphological Pathways Framework, to my knowledge, never mentions morphological knowledge. There are three separate concepts related to morphology in the framework: morphological awareness, morphemes, and morphology. They refer to different stuff and are not the same. Similarly, the Word Parts Levels Test measures affix knowledge sge. It is not the same as morphological knowledge. It is very crucial for the authors to give a clear definition of what is morphological knowledge discussed in this paper and constrain what they want to examine.
Line 131-135: The same as comments above. A clear definition of morphological knowledge is necessary. The Morphological Frameworks Pathways use morphological awareness, morphemes, and morphology in these three knowledge sources. I don’t think that they refer to the exact same concept used in this paper.
Line240-245 (Participants): Please provide more details regarding the participants’ demographic information, such as age, gender, language learning background, etc.
Line 246-260: I have a lot of issues with using this test to measure morphological knowledge. There are so many things going on in this test. The test designer mentioned that this test is used to measure affix knowledge as part of vocabulary knowledge. It is unclear how the three subsections represent different aspects of morphological knowledge.
Line 276-282: Citations needed.
Line 305-312 (Procedure): Did the participants have access to external resources, such as dictionary, online translator, and notes during the reading test? If so, how would this affect students’ performance, especially given that there was no time limit or proctor of this test?
Discussion: Given the lack of definition of morphological knowledge and what the three aspects really measure, it seems to me that the discussion needs some clarification, especially when discussing how they contribute to reading comprehension.
Comments on the Quality of English Language
I think the overall writing is good, though some proofreading will be helpful.
Author Response
Comments:
Overall, the study has a clear problem formulation. It clearly reviews the relevant literature and proposes a problem that needs more research attention. The writing is concise and logical. However, I have a few concerns regarding its definition of key terms and tasks adopted. Please see below for details:
> Thank you for the constructive comment. Our responses are shown by > after each comment.
Line 24-29: More citations needed.
> We added citations.
Line 64: [3]: I am not sure this is how to do in-text citation in the current citation format. It seems that Author Name [3] more commonly used in other articles published in this journal. The same comment applies to all similar in-text citations in this manuscript.
> Thank you for this comment. We will consult the editors.
Line 85-87: Morphological Pathways Framework, to my knowledge, never mentions morphological knowledge. There are three separate concepts related to morphology in the framework: morphological awareness, morphemes, and morphology. They refer to different stuff and are not the same. Similarly, the Word Parts Levels Test measures affix knowledge sge. It is not the same as morphological knowledge. It is very crucial for the authors to give a clear definition of what is morphological knowledge discussed in this paper and constrain what they want to examine.
> Thank you for this comment. We agree that “morphological awareness, morphemes, and morphology” in the MPF refer to different things. We changed “morphological knowledge” to “morphology” or “morphological awareness” as the model used these terms. We are not quite certain about the reviewer’s definition of “morphological knowledge”. We used the term “morphological knowledge” as a general term encompassing different types of knowledge and skills relating to morphology. We added this conceptualization in the Introduction. The focus of our study is derivational morphology or affix knowledge as the reviewer correctly mentioned. In our definition of the terms, affix knowledge is part of morphological knowledge. We changed morphological knowledge to affix knowledge when referring to our study throughout the paper.
Line 131-135: The same as comments above. A clear definition of morphological knowledge is necessary. The Morphological Frameworks Pathways use morphological awareness, morphemes, and morphology in these three knowledge sources. I don’t think that they refer to the exact same concept used in this paper.
> We first tried to revise according to the comment, but deleted these four lines in the end. We did not mean to equate “morphological awareness, morphemes, and morphology” in the MPR to “Form, Meaning, and Use” in our study. Including the Morphological Frameworks Pathways here may cause confusion among readers. Instead, we slightly modified a paragraph starting with “L2 researchers are also aware of…”. The reason is to give a little more emphasis on the theoretical base of “Form, Meaning, and Use”.
Line240-245 (Participants): Please provide more details regarding the participants’ demographic information, such as age, gender, language learning background, etc.
> We added gender, English learning background, and some other information. We did not report age because we did not ask them. For your information, first year university students in Japan are typically 18 to 19 years old, but could be older.
Line 246-260: I have a lot of issues with using this test to measure morphological knowledge. There are so many things going on in this test. The test designer mentioned that this test is used to measure affix knowledge as part of vocabulary knowledge. It is unclear how the three subsections represent different aspects of morphological knowledge.
> As we answered above, our concept of “morphological knowledge” is broad and includes affix knowledge. Reading research and vocabulary research are two related fields. It is not uncommon that morphology tests developed by vocabulary researchers are used in L2 reading research (e.g., Jeon, 2011), although such tests are not like morphological awareness measures commonly used in L1 research. However, we believe tests developed by vocabulary researchers also measure knowledge of morphology. Furthermore, when L2 researchers attempt to adopt morphology tests commonly used in L1 research, substantial changes are sometimes needed because of the difference in language background between L1 and L2 readers (e.g., Jeon, 2011). The reviewer’s comment, which is valid, seems to be related to difficulties underlying the research field of morphology and reading comprehension. We further expanded the discussion on the measurement issues in the final paragraph (limitations).
Line 276-282: Citations needed.
> Citations were added.
Line 305-312 (Procedure): Did the participants have access to external resources, such as dictionary, online translator, and notes during the reading test? If so, how would this affect students’ performance, especially given that there was no time limit or proctor of this test?
> No, participants did not access to external resources during the reading test (there was a time limit and proctor). Maybe, the reviewer meant the affix test. We added explanations in the procedure section and material section.
Discussion: Given the lack of definition of morphological knowledge and what the three aspects really measure, it seems to me that the discussion needs some clarification, especially when discussing how they contribute to reading comprehension.
> Conceptual and methodological issues raised by the reviewer are accommodated, as much as we can, throughout the paper (especially background and discussion sections). As another reviewer mentioned, the issue raised by this reviewer may not be constrained in this paper but is relevant to the problems in the field of morphology and reading. For instance, other researchers use the term “morphological knowledge” as we did in this paper. We believe we need more efforts to make advancements both in concepts and measurement in this research field.
Round 2
Reviewer 1 Report
Comments and Suggestions for Authors
The authors addressed all the comments/suggestions I made.
Reviewer 2 Report
Comments and Suggestions for Authors
Thank you so much for taking the time and addressing our concerns in our previous review. Your paper improved a lot with the changes you made in V2.0.